# Deformation Mechanism of Solidified Ti₃Al Alloys with Penta Twins under Shear Loading

**Xiaotian Guo** [1,*], **Han Xie** [2], **Zihao Meng** [1] **and Tinghong Gao** [2]

1   School of Physics and Electronic Engineering, Xinxiang University, Xinxiang 453003, China
2   Institute of New Optoelectronic Materials and Technology, College of Big Data and Information Engineering, Guizhou University, Guiyang 550025, China
*   Correspondence: guoxiaotian@xxu.edu.cn

**Abstract:** Owing to the excellent mechanical properties of the Ti₃Al alloy, the study of its microstructure has attracted the extensive attention of researchers. In this study, a Ti₃Al alloy was grown based on molecular dynamics using a decahedral precursor. Face centered cubic nanocrystals with tetrahedral shapes were formed and connected by twin boundaries (TBs) to form penta twins. To understand the shear response of the Ti₃Al alloy with multiple and penta twins, a shear load perpendicular to the Z-axis was applied to the quenched sample. The TBs slipped as Shockley dislocations commenced and propagated under shear loading, causing the detwinning of the penta twins and the failure of the system, indicating that the plastic deformation had been due to Shockley dislocations. The slip mechanism of multi-twinned structures in the Ti₃Al alloy is discussed in detail. This study would serve as a useful guide for the design and development of advanced alloy materials.

**Keywords:** TiAl alloys; induced solidification; shear response; slip system





## 1. Introduction

Ti–Al-based alloys have attracted extensive attention, owing to their low density [1], high strength [2], and high-temperature creep resistance [3]. Many studies on the structures and performance of Ti–Al-based alloys have been reported [4,5]. Considering that the formation of microstructures is closely related to solidification, Gao et al. have studied the solidification process [6,7] and mechanical response [8] of Ti–Al-based alloys solidified under high cooling rates, and penta-twinned structures were observed [9].

Penta-twinned material consists of five parts of face centered cubic (FCC) atoms separated by hexagonal close-packed (HCP) atoms as twin boundaries. Due to the low stacking fault energy in some metals, it is easy to produce penta twins in metal engineering processes such as annealing, strain, and rapid solidification. As a peculiar structure, penta twins improve the optical [10], electrical [11], and magnetic properties [12] of some nanocrystals; thus, it is an effective method for improving a material's performance. Many studies have been conducted to examine the mechanism by which penta twins influence the properties of various materials. As penta twins can enhance the mechanical properties of some materials [13,14], their effects on the mechanical properties of Ti₃Al alloys need to be investigated.

Recently, mechanical processes have been performed on Ti–Al-based alloys to study their mechanical properties and shear behavior [15,16], phenomena which deeply affect the structure of materials, and Ti₃Al alloys with penta twins would be no exception. It is necessary to study the shear response of Ti₃Al alloys with penta twins and clarify how penta twins influence the mechanical behavior of Ti₃Al alloys. Although several studies on penta twins have been reported, they have mainly focused on nanowires [17,18] and nanocrystalline alloys [19,20], whereas only a few have considered the shear response of penta twins. Chen et al. [21] studied the shear process and the deformation behavior of penta twins, but they explored only the deformation behavior of a single penta twin on the

surface, whereas the performance of penta twins in multiple twin systems is still difficult to predict, and some details about the evolution of the penta twins during deformation can be lost during experiments.

To obtain regular penta twins in this study, we employed precursor-induced growth based on molecular dynamics (MD) to obtain a solidification system, as induced growth of metals is an important means to control the structure of alloys, and the detailed evolution of penta twins can be obtained by MD simulations of shear processes. A decahedral precursor was introduced into the solidification process of the Ti$_3$Al alloy as a regular penta-twinned structure to induce the formation of a Ti$_3$Al alloy system containing multiple twins, and its shear process was simulated. The critical change point of the internal structure is found by average atomic potential energy, and Shockley dislocations are identified by the dislocation extraction algorithm (DXA) [22] in visualization software (the Open Visualization Tool, OVITO) [23]. The nucleation and propagation details of Shockley dislocations are also visualized by OVITO and, as a result of that, the deformation and failure mechanisms of the Ti$_3$Al alloy system with penta twins under shear stress were obtained.

## 2. Simulation Methodology

### 2.1. EAM Potential

The embedded-atom method (EAM) empirical potential function [24] was used to perform the induced quenching and shear process of Ti$_3$Al alloy simulated by MD because it has been widely used to model the solidification [25,26] and mechanical processes [27,28] of Ti–Al-based alloys. Over the past few decades, researchers have obtained numerous results using this function because it accurately reflects the interaction of Ti and Al atoms. The total energy of one system by EAM is expressed:

$$E_{total} = \frac{1}{2} \sum_{i,j} \Phi_{ij}(r_{ij}) + F_i(\overline{\rho_i}) \tag{1}$$

where $\Phi_{ij}$ denotes pair-interaction energy of atoms $i$ and $j$ at sites $\vec{r_i}$ and $\vec{r_j}$, $F_i$ represents the embedding energy of atom $i$, $\overline{\rho_i}$ is the host electron density and it is given by

$$\overline{\rho_i} = \sum_{i \neq j} \rho_i(r_{ij}) \tag{2}$$

For the system of Ti$_3$Al, make the following conversions:

$$\rho_{Ti} \rightarrow s_{Ti}\rho_{Ti}(r) \tag{3}$$

$$\rho_{Al} \rightarrow s_{Al}\rho_{Al}(r) \tag{4}$$

$$F_{Ti}(\overline{\rho}) \rightarrow F_{Ti}\left[\frac{\rho_{Ti}(r)}{s_{Ti}}\right] \tag{5}$$

$$F_{Al}(\overline{\rho}) \rightarrow F_{Al}\left[\frac{\rho_{Al}(r)}{s_{Al}}\right] \tag{6}$$

$$F_{Ti}(\overline{\rho}) \rightarrow F_{Ti}(\overline{\rho}) + g_{Ti}\overline{\rho} \tag{7}$$

$$F_{Al}(\overline{\rho}) \rightarrow F_{Al}(\overline{\rho}) + g_{Al}\overline{\rho} \tag{8}$$

$$\Phi_{TiTi}(r) \rightarrow \Phi_{TiTi}(r) - 2g_{Ti}\rho_{Ti}(r) \tag{9}$$

$$\Phi_{AlAl}(r) \rightarrow \Phi_{AlAl}(r) - 2g_{Al}\rho_{Al}(r) \tag{10}$$

where $s_{Ti}$, $s_{Al}$ are constant term, $g_{Ti} = -F'_{Ti}(\overline{\rho_{Ti}^0})$, $g_{Al} = -F'_{Al}(\overline{\rho_{Al}^0})$, where $\overline{\rho_{Ti}^0}$ and $\overline{\rho_{Al}^0}$ are the equilibrium electron densities of atoms.

### 2.2. Simulation Details

A 130 nm × 130 nm × 130 nm box with 107,300 atoms (80,475 Ti and 26,825 Al atoms) was constructed. To introduce regular penta twins, the induced solidification with a decahedral precursor of $Ti_3Al$ alloy (Figure 1a) was built in center of the box, and the area around the precursor was filled with Ti and Al atoms in random positions. Figure 1b is the structure of the truncated decahedron, which is widely evident in metal nanocrystals, and serves as the core structure of the penta twin [29]. The quenched system was used to perform a shear process to obtain the deformation and failure mechanisms of the $Ti_3Al$ alloy with penta twins. All of the surfaces of the box were set to be periodic. The thermodynamic process was simulated under the NPT (constant atom number, pressure, and temperature) ensemble. The temperature was controlled at 2000 K using a Nose/Hoover temperature thermostat, and it ran for 500 ps with a time step of 1 fs to perform a relaxation process. Next, the cooling from 2000 to 200 K was simulated at a quenching rate of $10^{11}$ K/s. Then, the quenched system was equilibrated at room temperature (300 K) for 300 ps to make it fully relaxed. Finally, a shear load was applied to the alloy system at a strain rate of $10^9$ s$^{-1}$. The whole simulation process is performed within large-scale atomic/molecular massively parallel simulator (LAMMPS) software [30].

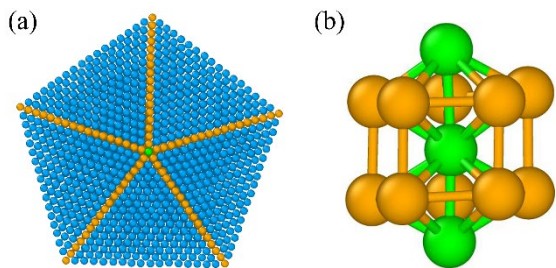

**Figure 1.** (**a**) Top view of the initial penta-twinned structure. (**b**) Truncated decahedron structures (TDH) colored green. The FCC structures are colored blue, and the HCP structures are colored orange.

## 3. Results

### 3.1. Induced Solidification

Analyzing the fluctuation in average atomic potential energy is crucial to understanding the change in the internal structure of materials during the quenching of the $Ti_3Al$ alloy. As shown in Figure 2a, the average atomic potential energy drops abruptly at 1345 K, and then resumes its linear decline when the temperature drops to 1252 K, indicating that the system crystallization process occurs mainly in this interval. Figure 2b shows the visualization of the system at different temperatures during solidification. Liquid atoms nucleate on the surface of the decahedral precursor at first, indicating heterogeneous solidification. Then, the rest of the liquids solidify on the initial nucleated sites with a decrease in temperature, and atoms of the crystal continuously expand around the initial penta twins in the temperature range of 1345–1252 K, indicating the formation of crystalline FCC and HCP structures. The sudden change in the average atomic energy curve corresponds to the fast increase in the abovementioned structures. In addition, many quenched twins are observed in the system.

Just as in the initial structure of the penta-twinned structure (Figure 1a), HCP structures serve as twin boundaries (TBs) to connect the crystalline FCC structures. Most FCC grains are nanocrystals with tetrahedral shapes, and their surfaces are crystalline FCC {111} planes. There are a considerable number of TDH structures (Figures 1b and 3a) around the initial penta-twinned structure after the crystalline FCC and HCP structures around the initial penta-twinned structure have been discarded, indicating the formation of a considerable number and size of penta twins in the alloy system. Thus, the details of these penta twins were visualized from the axial direction (Figure 3b), in which HCP and TDH structures were eliminated for clear observation. Many tetrahedral nanocrystals similar in size to

the initial one (the black circle in Figure 3b) are formed under the induction of the initial penta-twinned structure, and some new penta twins are obtained.

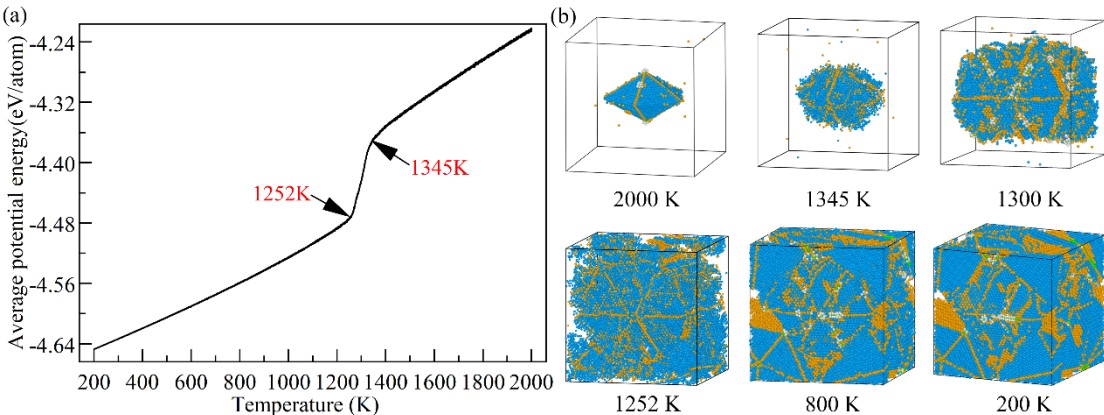

**Figure 2.** (**a**) Average atomic energy and (**b**) snapshots of several temperature points. Atoms detected as others by OVITO were removed. FCC structures are colored blue, and HCP structures are colored orange.

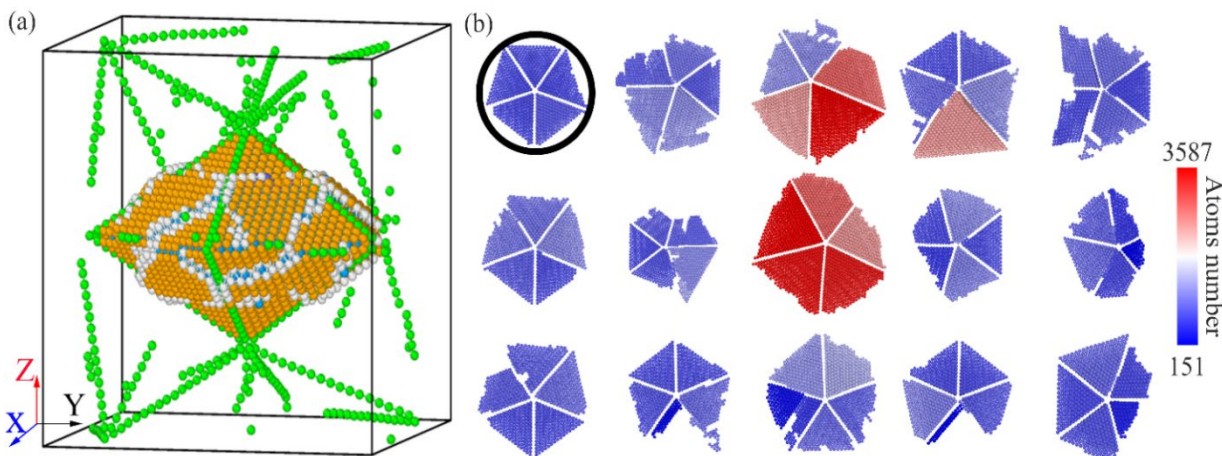

**Figure 3.** (**a**) TDH atoms (green) around the initial penta-twinned structure and the multi-twinned regions formed by FCC tetrahedron structures. Atoms of other structures are marked white. Atoms in (**b**) are colored based on the atom number of each cluster.

### 3.2. Shear Characteristics

To study the shear behavior during the mechanical process of Ti–Al-based alloys, shear loading in a direction perpendicular to the Z-axis, which deviates from the axial direction of the initial penta-twinned structure, was applied to examine the mechanical behavior of the Ti$_3$Al alloy with multiple twins. The stress-strain response is shown in Figure 4a. Before point ①, the stress-strain curve rises in an almost straight line, and this phase follows the elastic stage of the classical Hooke's law. When point ① is crossed, the material begins to enter the yielding stage, and the linear relationship between stress and strain is broken, with point ② being the upper yielding point and point ③ the lower yielding point. The point ③–④ stage is the strengthening stage, and this stage increases with the increase of strain. After reaching point ④, the material produces a local deformation, and then crosses point ⑤ and enters the plastic flow stage. The shear response of the induced system is divided into elastic-deformation (stage A) and plastic-flow stages (stage B). Before the strain of point ④, a high level of shear stress is obtained, and the fluctuation in the curve shows strain-hardening effects during sample deformation. Considering that

dislocation behavior can influence the mechanical process of the system, the changes in the dislocation length were evaluated (Figure 4b). The dislocation density in the system increases rapidly (the total length of dislocations increases, whereas the volume of the system is almost unchanged) at the high-shear-stress stage, as shown in the dislocation length curve. Moreover, the corresponding snapshots in Figure 4a at the key points show significant slips in the system, slips which contribute to stress relaxation in the quenched sample. The rapid drop in the stress-strain curve from points ④ to ⑤ is contributed by the slip of the overall system, as the stress needed to deform the sample in the subsequent deformation is reduced by the accumulation of dislocations under high-level shear stress; at this stage the material has developed a large local deformation.

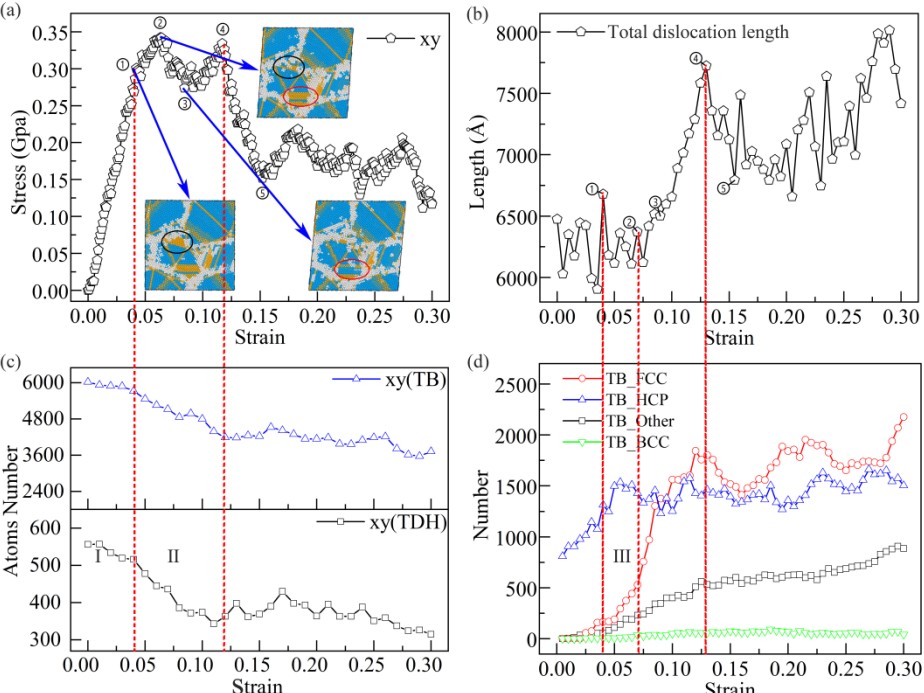

**Figure 4.** (**a**) Stress-strain curve and snapshots of some strain points. Changes in (**b**) dislocations, (**c**) TB atoms and TDH atoms, and (**d**) atoms transformed from TB to various structures. Stage A: elastic-deformation stage; Stage B: high-level stress stage.

Figure 4c shows the changes in TB and TDH atoms during deformation. TB and TDH atoms are continuously lost during deformation, and a dynamic equilibrium is eventually reached, indicating that the induced penta twins are closely related to the slip behavior in the system. To understand the relationship between penta twins and the slip behavior, the transformation from TB atoms to other types of atoms (FCC, HCP, Others, and BCC, as defined in OVITO) is shown in Figure 4d, where TB atoms are selected by the initial state of the quenched sample. Several TB atoms transform into HCP structures at the elastic-deformation stage, because the surrounding atomic environment of TB atoms is changed by local adjustments of atoms caused by elastic strain in the system. Then, TBs slip with an increase in strain during deformation because TB atoms are influenced by the surrounding atomic environment, and begin to slip significantly. The transformation from TB to the FCC structure indicates the slip of TBs (Figure 4d; from points ① to ④). However, further characterization of the evolution of the initial penta-twinned structure (Figure 5) shows that, though the TBs slip, the penta-twinned structure still exists in the system with a continuous increase in strain. Therefore, to determine the plastic deformation mechanism during the shear process, further characterization of the shear deformation requires more details of the dislocations and microstructures during deformation.

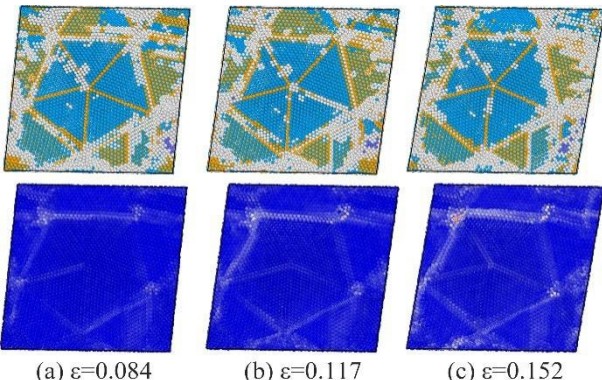

(a) ε=0.084      (b) ε=0.117      (c) ε=0.152

**Figure 5.** Snapshots of several strain points. Atoms are colored based on both the structure type and atomic strain of each atom. FCC structures are colored blue, and HCP structures are colored orange, other atoms are colored gray. Where ε is the strain rate, (**a**) ε = 0.084, (**b**) ε = 0.117, (**c**) ε = 0.152.

### 3.3. Plastic Deformation Due to the Interaction between Dislocations and TBs

Figures 6–8 show detailed snapshots of dislocations and micro-structures of the TBs during deformation. In the quenched sample, tetrahedron nanocrystals are connected by TBs, forming both regular penta and other twinned regions with many defects. At the elastic-deformation stage, the local structure can be adjusted much more easily in other twinned regions than in the regular penta-twinned regions. As is shown in Figure 6, the Shockley dislocations (the black circle) intersected with TB and propagated towards both sides, forming complex stacking faults. It is much easier for Shockley dislocations to nucleate and propagate in the defective multi-twinned regions with the splitting axis in the initial state and Shockley dislocations along the TB, as shown in Figure 7. Herein, defective TBs are formed by the nucleation of Shockley dislocations (circled by black solid line) at TB, and the evolution of defective TBs originates from the propagation of Shockley dislocations at TBs. Interestingly, the splitting axis intersects at one point because of the TB slip induced by the propagation of Shockley dislocations during the continuous deformation of the defective multi-twinned region, whereas the regular penta-twinned regions can retain the basic structure (from Figure 8a to Figure 8b). Meanwhile, the nucleation and propagation of Shockley dislocations are activated on TBs, and they serve as the surfaces (circled by a red dot line in Figure 8b) of the penta-twinned structure. Further TB slip inside the penta-twinned structure is activated with the development of shear strain. Details of the propagation of dislocations outside and inside the penta-twinned structure are shown in Figure 8. The penta-twinned structure deforms from outside to inside. The internal structures of the penta-twinned structure are affected by the accumulation of deformations and show as TB slip, causing the detwinning of the penta-twinned structure. Eventually, large deformation of the overall system occurs, resulting in material failure.

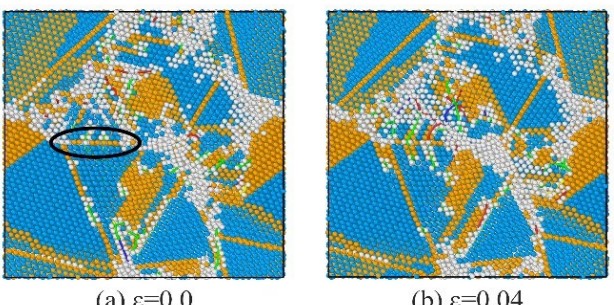

(a) ε=0.0      (b) ε=0.04

**Figure 6.** Slices of the evolution of dislocations in defective TBs at the initial stage of shear loading. FCC structures are colored blue, and HCP structures are colored orange, other atoms are colored gray. (**a**) ε = 0.0, (**b**) ε = 0.04.

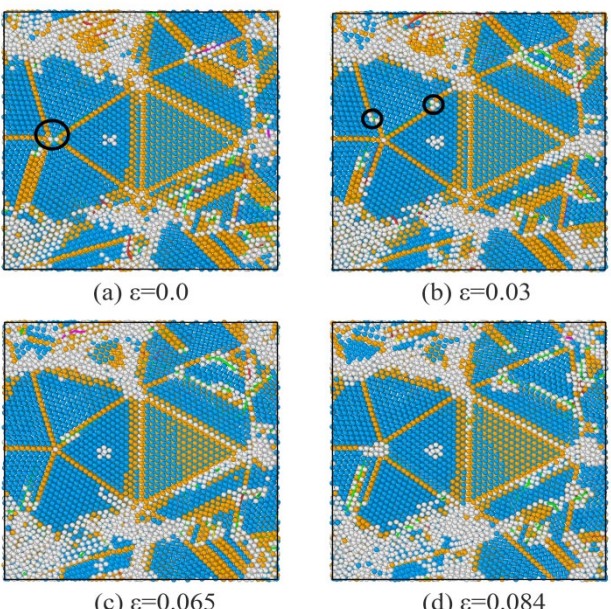

**Figure 7.** Slices of the evolution of dislocations in a defective penta-twinned region. FCC structures are colored blue, and HCP structures are colored orange, other atoms are colored gray. (**a**) $\varepsilon = 0.0$, (**b**) $\varepsilon = 0.03$, (**c**) $\varepsilon = 0.065$, (**d**) $\varepsilon = 0.084$.

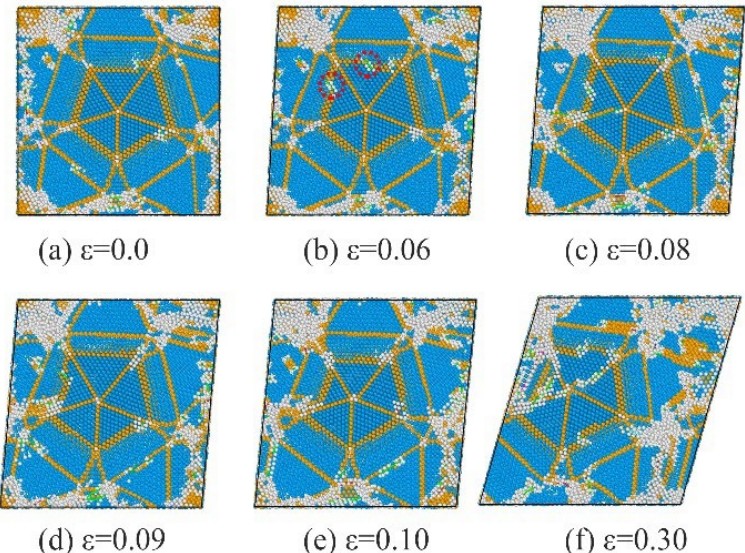

**Figure 8.** Slices of the evolutions of dislocations in the regular penta-twinned region. FCC structures are colored blue, and HCP structures are colored orange, other atoms are colored gray. (**a**) $\varepsilon = 0.0$, (**b**) $\varepsilon = 0.06$, (**c**) $\varepsilon = 0.08$, (**d**) $\varepsilon = 0.09$, (**e**) $\varepsilon = 0.10$, (**f**) $\varepsilon = 0.30$.

## 4. Discussion

From the above analysis, nucleation and propagation of Shockley dislocation occur throughout the shear-deformation stage, which is indicated by the total length of dislocations in each multi-twinned region (Figure 9). The total length of Shockley dislocations in each multi-twinned region is significantly higher than that of other dislocations, indicating that the structural changes in multi-twinned regions are dominated by the evolution of Shockley dislocations. To determine the mechanism by which the nucleation and propagation of Shockley dislocations affect the deformation of the penta-twinned structure, several snapshots of some key points in the deformation of penta-twinned structure are

shown in Figure 10. Atoms detected as FCC structure at first were eliminated for a better representation.

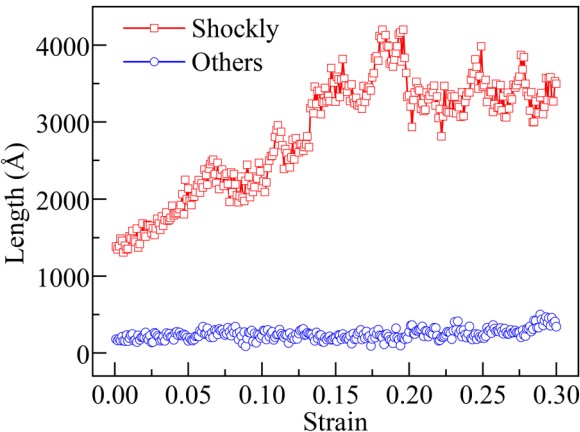

**Figure 9.** Sum of the lengths of dislocations (Shockley dislocations and other types) in independent multi-twinned regions. FCC structures are colored blue, and HCP structures are colored orange, other atoms are colored gray.

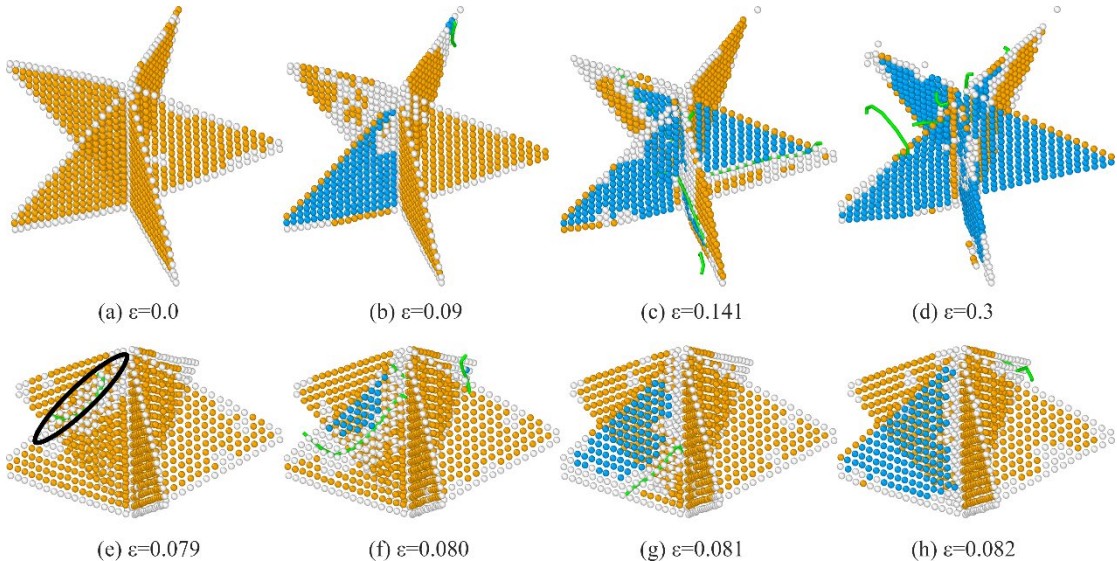

**Figure 10.** Snapshots of the slippage contributed by the propagation of Shockley partial dislocations in the TB of penta twins: (**a**–**d**) Evolutions of slips in each TB; (**e**–**h**) Propagation of Shockley dislocations on the first TB. FCC structures are colored blue, and HCP structures are colored orange. (**a**) $\varepsilon = 0.0$, (**b**) $\varepsilon = 0.09$, (**c**) $\varepsilon = 0.141$, (**d**) $\varepsilon = 0.3$, (**e**) $\varepsilon = 0.079$, (**f**) $\varepsilon = 0.080$, (**g**) $\varepsilon = 0.081$, (**h**) $\varepsilon = 0.082$.

As shown in Figure 10e, the Shockley dislocations (circled by a black solid line) nucleate and propagate on the TBs, causing the local adjustment of TBs and HCP–FCC transformation. Almost all atoms in the single TB (Figure 10a) are transformed into the FCC structure when the Shockley dislocation extends to the bottom of the TB and greatly damages the penta axis, indicating a full slip of the single TB, which contributes to the split of the penta axis (Figure 8), which, in turn, facilitates the propagation of Shockley dislocations nucleated from the penta axis on other TBs, and eventually causes the failure of the penta-twinned structure. TBs and crystalline structures slip in the basal slip system, represented as $\{0\,0\,0\,1\} <-2\,1\,1\,0>$, as shown in Figure 11. However, the slip path in TBs is divided into two steps: at first, the $\{0\,0\,0\,1\}$ crystalline plane slips along $<-1\,1\,0\,0>$, and then along $<-1\,0\,1\,0>$, as shown in Figure 12. The relative positions of the TB atoms and

red atoms do not change during the two steps of the slip path shown in Figure 12, whereas the TB atoms are temporarily identified as FCC structure (sky blue atoms) due to the slip of the dark blue atoms. The TB atoms are then identified as HCP structure (orange atoms) again in the subsequent slip process of the dark blue atoms.

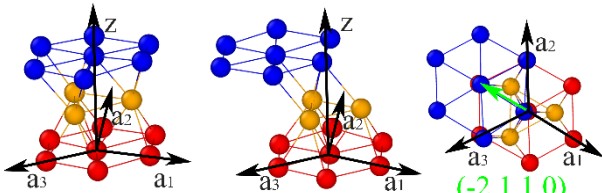

**Figure 11.** Basal slip system in penta twins.

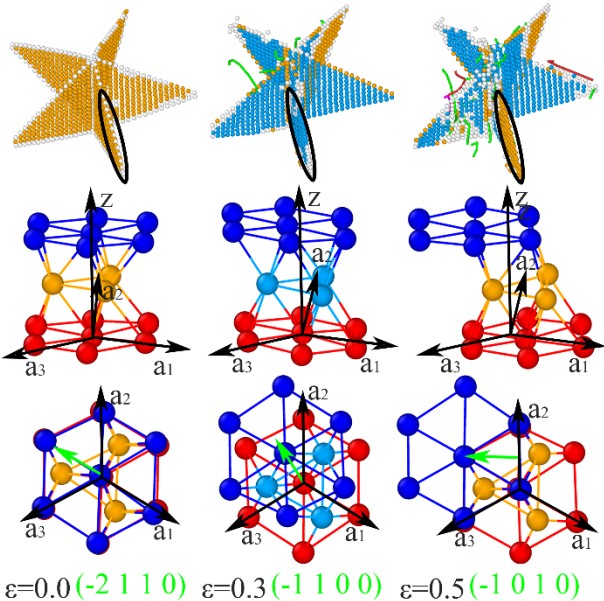

**Figure 12.** The basal slip system is divided into two steps in TBs in penta twins. The atoms around the TB atoms are colored dark blue and red.

## 5. Conclusions

In this work, MD simulations were employed to investigate the induced solidification and shear response of a $Ti_3Al$ alloy with penta twins. Based on the results, the following conclusions are drawn:

(1) A sample with multiple twins can be obtained by induced solidification using penta-twinned structures under a quenching rate of $1.0 \times 10^{11}$ K/s, during which induced penta twins are formed by FCC nanocrystals with tetrahedral shapes, and defective multitwinned regions can also be observed.

(2) Dislocations nucleate and propagate easier in defective multitwinned regions than in regions without defects, as defects reduce the stress needed to deform a sample. Correspondingly, defects outside the regular penta twins enhance the nucleation and propagation of dislocations on TBs serving as the surfaces of the regular penta twins. Thus, dislocations nucleate and further propagate on TBs in the penta twins, causing the failure of the penta twins.

(3) The slip system of a $Ti_3Al$ alloy with multitwinned structures is the $\{0\,0\,0\,1\} <-2\,1\,1\,0>$ basal slip under shear loading, and there are obvious strain-hardening effects during deformation. The slip path in TBs is divided into two steps: the $\{0\,0\,0\,1\}$ crystalline plane slips along $<-1\,1\,0\,0>$ first and then along $<-1\,0\,1\,0>$.

(4) Interactions between Shockley dislocations and TBs cause the migration of TB and the failure of penta twins, which could be the mechanism of plastic deformation in Ti₃Al alloys with penta twins.

**Author Contributions:** Conceptualization, X.G. and T.G.; methodology, X.G. and T.G.; software, H.X.; validation, T.G. and X.G.; formal analysis, T.G., X.G and H.X.; investigation, H.X. and Z.M.; resources, T.G.; data curation, H.X. and Z.M.; writing—original draft preparation, H.X.; writing—review and editing, T.G. and X.G.; visualization, H.X. and Z.M.; supervision, X.G. and T.G.; project administration, H.X.; funding acquisition, T.G. and X.G. All authors have read and agreed to the published version of the manuscript.

**Funding:** This research was funded by the Industry and Education Combination Innovation Platform of Intelligent Manufacturing and Graduate Joint Training Base at Guizhou University (Grant No: 2020-520000-83-01-324061), the National Natural Science Foundation of China (Grant nos.51761004), the Guizhou Province Science and Technology Fund (Grant no. ZK[2021] 051), and the Ninth Group of Key Disciplines in Henan Province (grant no. 2018119).

**Institutional Review Board Statement:** Not applicable.

**Informed Consent Statement:** Not applicable.

**Data Availability Statement:** Not applicable.

**Conflicts of Interest:** The authors declare no conflict of interest.

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
