# Peer review of "Deformation Mechanism of Solidified Ti3Al Alloys with Penta Twins under Shear Loading"

_metals, doi:10.3390/met12081356_

Round 1

Reviewer 1 Report

Previously, the authors have shown formation of penta-twins in Ti3Al alloys by simulating solidification by computation. These penta-twins are interesting atomic structures with significance for deformation mechanism of these alloys. In this report, they show effect  of simulated shear loading on deformation mechanism. These can be of great interest to some readers.

The introduction should introduce more clearly to the readers as to what are penta-twins, what  is its geometry and atomic structure. And why they are formed. What is a decahedron, and truncated decahedron? Term TDH is introduced only in the figure caption and not  in the text. Besides, full form of TB (and also NPT) is never written.

Fig 2 or 2a is not referred before 2b. z-axis is not marked in Fig. 3a. Fig. 3a: the green atoms are TDH or HCP? 

"rapid drop in stress ... as the stress needed to deform ... is reduced by accumulation of dislocations": this is counter-intuitive to strain hardening!

Fig. 4a: two types of lines should be distinguished: delineating the strain stages, and pointing to the simulated images (can be arrows). Y-axis in Fig. 4b should be written as total dislocation length, rather than just Length. 

line 149: Fig. 4d should be 4a?

Fig. 10 caption: what do atom colors represent?

line 203: duplicate 'shown in' should be deleted.

line 245: 'Please add' is supposed to be deleted?

Fig. 12 should be explained in more detail.

In general, Discussion could be more comprehensive.

Reviewer 2 Report

The creation of alloys is the most promising method for increasing the strength of materials by controlling the interaction of dislocations and defects in the crystal structure during plastic deformation. Interest in the study of Ti3Al alloys remains traditionally high, since they combine relative low weight and fairly high strength characteristics. Since penta twins can improve the mechanical properties of alloys, the study of their effect on the mechanical properties of Ti-Al alloys is of interest from an academic and practical point of view.

The peer-review paper is devoted to MD simulations the induced solidification and shear response of a Ti3Al alloy with penta twins.

As a result of the investigation, the authors established and formulated a number of regularities.

·         A sample with multiple twins can be obtained by  induced solidification using penta-twinned structures under a quenching rate of 1011 K/s .

·         Dislocations nucleate and propagate easier in defective multitwinned regions  than in regions without defects.

·         The defects outside the regular penta twins enhance the nucleation and  propagation of dislocations on TBs serving as the surfaces of the regular penta twins.

·         Interactions between Shockley dislocations and TBs cause the migration of TB and  the failure of penta twins, which could be the mechanism of plastic deformation in Ti3Al alloys with penta twins.

This study provides a theoretical basis and would serve as a useful guide for the design and development of advanced alloy materials.

The article contains a number of interesting results that may be useful to a potential reader. The quality of the presentation of the material of the article, from the point of view of the reviewer, is good. The article may be published after minor editing.

There are several questions, comments and suggestions for the authors.

1.       The authors are recommended to pay more attention to the description of the details of mathematical modeling. In particular, it would be more convenient for a potential reader to evaluate the presented results if the empirical potential function for atomistic simulations of the Ti-Al system were given in the text of the article.

2.       Figure 4 shows 5 characteristic points. However, a detailed judgment is made only for sections 4 - 5. From the point of view of the reviewer, the authors should, at least briefly, characterize sections from 0 to 1 (linear elasticity obviously takes place here), as well as the processes occurring in sections 1-2, 2-3, 3-4 and to the right of point 5.

3.       On fig. 9 represents the total length of Shockley dislocations. This dependence has a rather complex character: an increase at small deformations, then a decrease and reaching constant values. How do the authors interpret this behavior?

4.       There are no references to figures 1b and 2a in the text of the paper.

5.       According to the reviewer, there is an error in the description of the vertical axis in Fig. 9.

6.       According to the reviewer opinion, there are typos in lines 77-80. The correct spelling of the text should look like: "Next, the cooling from 2000 to 200K was simulated at a quenching rate of 1011 K/s. Then, the quenched system was equilibrated at room temperature (300 K) for 300 ps to make it fully relaxed. Finally, a shear load was applied to the alloy system at a strain rate of 109 s-1”.

7.       According to the reviewer opinion, there are typos in lines 222-225. The correct spelling of the text should look like: “(1) A sample with multiple twins can be obtained by induced solidification using penta-twinned structures under a quenching rate of 1.0×1011K/s, during which induced penta twins are formed by FCC nanocrystals with tetrahedral shapes, and defective multitwinned regions can also be observed.”
